# Conversion of Oil-Containing Residue from Waste Oil Recycling Plant into Porous Carbon Materials Through Activation Method with Phosphoric Acid

**DOI:** 10.3390/ma17246161

**Published:** 2024-12-17

**Authors:** Li-An Kuo, Wen-Tien Tsai, Chien-Chen Pan, Ya-Chen Ye, Chi-Hung Tsai

**Affiliations:** 1Department of Environmental Science and Engineering, National Pingtung University of Science and Technology, Neipu, Pingtung 912, Taiwan; sanck112204@gmail.com; 2Graduate Institute of Bioresources, National Pingtung University of Science and Technology, Neipu, Pingtung 912, Taiwan; yeyachen1026@gmail.com; 3Department of Environmental Engineering and Science, National Pingtung University of Science and Technology, Neipu, Pingtung 912, Taiwan; joepan@dejing-eco.com; 4De-Jing Enterprise Co., Neipu, Pingtung 912, Taiwan; 5Department of Resources Engineering, National Cheng Kung University, Tainan 701, Taiwan; ap29fp@gmail.com

**Keywords:** oily sludge, chemical activation, phosphoric acid, washing modification, pore analysis

## Abstract

In the waste oil recycling industry, large amounts of oil-containing sludge are still generated, thus posing a resource depletion issue when disposed of or incinerated without energy recovery or residual oil utilization. In this work, chemical activation experiments using phosphoric acid (H_3_PO_4_) were performed at a low temperature (600 °C) for 30 min to produce porous carbon products. From the results of the pore property analysis, an increasing trend with an increasing impregnation ratio from 0.5 to 2.0 was observed. Based on the Brunauer–Emmett–Teller (BET) model, the maximal BET surface area was about 70 m^2^/g, which was indicative of the hysteresis loop and the type IV isotherms in the resulting carbon product. In addition, the enhancement in the pore properties of the carbon products obtained through acid-washing was superior to that achieved through water-washing and without post-washing. From observations made using scanning electron microscopy (SEM) and energy-dispersive X-ray spectroscopy (EDS), the carbon products featured a porous structure and inherent magnetism due to their richness of iron oxides. In this regard, they can be used as efficient adsorbents or catalyst supports due to their simple recovery (or separation) when exhausted.

## 1. Introduction

With the rapid development of industrial and commercial activities and the increase in the population, energy demand and environmental issues are becoming more and more serious. Recycling all kinds of waste oil has attracted significant attention in recent years [1]. These oil sources include engine oil (or mineral engine oil), lubricating oil, cutting oil, tank-bottom oil, and so on. For example, so-called waste engine oil refers to engine oil that has been mixed with water, dust, and metal-containing powder produced from mechanical wear and tear like iron (Fe), lead (Pb), zinc (Zn), and magnesium (Mg) [2]. Without waste oil collection and reuse by recycling it into oil-related or oil-derived products, it can pose adverse environmental and health impacts due to its contents of petroleum-based components, heavy metals, and other contaminants [3]. Although the production of alternative fuels from waste oils can reduce the consumption of petroleum-based fuels and thus reduce greenhouse gas (GHG) emissions [4], the waste oil refining process still generates large amounts of oil-containing residue. Based on the current situation, the conversion of oil-containing residue from waste oil recycling plants into valuable products not only provides full solutions to issues in the waste oil recycling industry but also creates additional profits for recycling companies.

In recent years, many studies have reviewed different thermal, chemical, and physical technologies to convert oil-containing residue or oily sludge into a variety of products [5,6,7,8,9]. In brief, these studies have focused on the recovery of oils (or residual oils) through the use of solvent extraction and ultrasonication and energy recovery through thermal treatment and thermochemical processes like pyrolysis and gasification. Regarding the production of porous carbon-rich materials from oily sludge through activation processes, only a few studies have been performed on waste oil recycling plants [10,11], while others have focused on oil (petroleum) refinery plants like oil tanks (oil at the bottom of tanks) [12,13,14,15,16,17,18,19,20,21,22,23,24,25,26,27,28]. Furthermore, most of these studies carried out chemical activation by using potassium hydroxide (KOH), while fewer used zinc chloride (ZnCl_2_). Exceptionally, Li et al. prepared de-ashed char at 700 °C and carried out post-washing with hydrogen fluoride (HF) solution for 24 h to remove inorganic impurities [18]. The authors used phosphoric acid (H_3_PO_4_) as an activating agent to promote the pore properties of the above-mentioned resulting char with different H_3_PO_4_/char ratios of 2:1, 2.5:1, and 3:1 at 400 °C for 1 h under a nitrogen flow [18]. To obtain highly porous carbon products, the crude products were washed with plenty of deionized (D.I.) water and then dried at 105 °C for 24 h. In the previous studies [10,11], the pore properties of the resulting carbon products prepared from oil-containing residue through the physical activation of carbon dioxide (CO_2_) at 850 °C for 0.5–1.5 h showed a decreasing trend [11]. The maximal Brunauer–Emmett–Teller (BET) surface area was only 21.59 m^2^/g. In addition, the data on the BET surface area were enhanced to 40.53 m^2^/g due to the acid-washing of the crude carbon products [10].

Based on the results of previous studies [10,11], a feasible approach to preparing carbon products from the residual sludge of a waste oil recycling plant was developed, even though the pore properties obtained were not significant. In this work, we carried out chemical activation experiments by using H_3_PO_4_ at a low temperature (600 °C). To enhance the pore properties by removing residual H_3_PO_4_ and inherent inorganic minerals, post-washing of the crude carbon products was performed by using deionized water and dilute acid. In this regard, the pore properties of the resulting carbon products were determined from the surface area and using a porosity analyzer based on nitrogen isotherms (−196 °C). The porous textures and elemental compositions on the surface were viewed with scanning electron microscopy (SEM) and energy-dispersive X-ray spectroscopy (EDS), respectively.

## 2. Materials and Methods

### 2.1. Materials

The initial precursor for producing the carbon products was derived from a waste oil recycling plant located in Neipu Industrial Park (Pingtung County, Taiwan). According to its official definition, as-synthesized oil-containing sludge in the form of dark and sticky slurry is a non-hazardous waste. The sample was analyzed to obtain information about its thermochemical data for proximate analysis, calorific value, elemental content, and thermogravimetric analysis (TGA). To perform the chemical activation experiments with phosphoric acid (or *ortho*-phosphoric acid), the concentration of the ACS-grade activation agent from Merck KGaA (Darmstadt, Germany) used was 85 wt%.

### 2.2. Thermochemical Properties of Oil-Containing Sludge

Baseline information about the thermochemical properties (including proximate analysis, calorific value, and TGA) of the oil-containing sludge was first determined to obtain its thermal decomposition behavior and potential for preparing carbon materials. Based on official standard methods (i.e., “Standard Method for Determining Moisture Content of Industrial Waste” and “Standard Method for Determining Combustible and Ash Contents of Industrial Waste”, coded as NIEA R203.02C and NIEA R205.01C, respectively) in Taiwan, a proximate analysis of the as-synthesized sludge sample was performed to assess the quality of the precursor. Aside from the determination of its moisture and ash contents, the sample was also found to be combustible. Therefore, a measure of the solid combustible fraction was obtained. The calorific value is the energy content per unit mass of the sample (dry basis), which was indicative of the heat energy released while it was completely burned, obtained by an adiabatic bomb calorimeter (Model: CALORIMETER ASSY 6200; Parr Co., Moline, IL, USA) in triplicate. It should be noted that the determined value represents the gross calorific value or higher calorific value because it contains the latent heat of combustion and the latent heat of water vaporization. On the other hand, the TGA was performed with a precision instrument (Model: TGA-51; Shimadzu Co., Kyoto, Japan). The dried sludge sample was uniformly heated at four rates (5, 10, 15, and 20 °C/min) under nitrogen gas (flowrate set to 50 cm^3^/min) up to 900 °C. In addition, the main elemental contents (including carbon, oxygen, alkali metals, alkaline earth metals, silicon, and other main elements) on the surface of the dried sludge sample were determined by using energy-dispersive X-ray spectroscopy (Model: 7021-H; HORIBA Co., Kyoto, Japan).

### 2.3. Phosphoric Acid Activation Experiments

Chemical activation is a low-energy-consumption process for producing porous carbon materials because it is generally performed at lower temperatures (400–800 °C) as compared to physical activation processes, which tend to be performed at higher temperatures (700–1000 °C) [29]. In the former process, phosphoric acid is a commonly used activator acting as a dehydrating agent. The preliminary data on the pore properties of the resulting carbon products prepared at 600–900 °C for 30–60 min with a high impregnation ratio of 2.0 (the mass ratio of phosphoric acid to oil-containing sludge) only showed non-regular variations in the specific surface area (BET model) ranging from 50.7 to 59.3 m^2^/g. Therefore, the chemical activation experiments were designed at a lower temperature (600 °C) with different impregnation ratios (i.e., 0.0, 0.5, 1.0, 1.5, and 2.0) in the present study. To carbonize the residual sludge, an inert atmosphere was created by purging nitrogen gas with a flowrate of 500 cm^3^/g during all the activation experiments. In order to evaluate the enhancement of the pore properties for the resulting carbon products, a post-washing treatment with deionized water and dilute acid solution (0.25 M HCl) was performed on a hot-plate at about 75 °C for 30 min. For the detailed procedures, one can refer to the previous studies [11,30,31]. In this work, the resulting carbon products were given specific codes. For example, C-05-SL-600-30, C-05-SL-600-30-W, and C-05-SL-600-30-A represent the carbon products (C) prepared from the sludge (SL) at 600 °C for 30 min without washing, with water-washing, and with acid-washing, respectively. To reduce the use of phosphoric acid, two acid-washing experiments with impregnation ratios of 0.0 and 0.5 were carried out in the present study. That is, the resulting carbon products were C-0-SL-600-30-A and C-05-SL-600-30-A.

### 2.4. Characterization of Carbon Products

The nitrogen adsorption/desorption isotherm at −196 °C is commonly used for determining the pore properties of a carbon material (or product), including its specific surface area, pore volume, average pore size, and pore size distribution [32,33]. An accelerated surface area and porosimeter (Model: ASAP 2020; Micromeritics Instrument Co., Norcross, GA, USA) was adopted for all pore property analyses. Prior to the determination of pore properties, degassing was performed under vacuum conditions (1.33 Pa) at a high temperature (200 °C) for 10 h. Regarding the specific surface area, the Brunauer–Emmett–Teller (BET) surface area (S_BET_) is most commonly used, which is expressed in units of area per mass of sample (m^2^/g). Through comparison with the adsorption isotherm of a nonporous reference material, data on micropore (pore size ˂ 2.0 nm) volume (V_micro_) and micropore surface area (S_micro_) can be calculated by using the “*t*-plot” method [33]. Herein, the *t*-plot method was used to estimate the surface area and pore volume with micropores (pore diameter or width: ˂2.0 nm) based on the Harkins–Jura equation. The data on total pore volume (V_t_) were estimated by assuming that all pores were filled with the liquid adsorbate (i.e., nitrogen). Therefore, the total nitrogen gas adsorbed at a saturated relative pressure (0.995) was converted to liquid nitrogen volume by using its density at −196 °C (i.e., 0.807 cm^3^/g). By assuming uniform cylindrical pores, the following equation was generally used to calculate the average pore diameter (D_av_) of a porous sample for a given total pore volume and BET surface area:D_av_ = (V_t_ ∗ 4)/S_BET_

On the other hand, the pore size distribution of a porous material was calculated based on the Barrett–Joyner–Halenda (BJH) method, which can be represented both in differential and cumulative models [33]. It should be noted that the BJH method was limited to displaying pore size distributions in the range of mesopores (2.0–50.0 nm) to macropores (>50.0 nm). In order to image the porous structures of the resulting carbon products, scanning electron microscopy (SEM) (Model: S-3000N; Hitachi Co., Tokyo, Japan) was applied under an accelerating voltage of 15.0 kV. Prior to the SEM observations, the non-conductive carbon products were coated with a gold film in an ion sputter (Model: E1010; Hitachi Co., Tokyo, Japan).

## 3. Results

### 3.1. Thermochemical Characterization of Oil-Containing Sludge

Table 1 lists data on the proximate analysis and calorific value of oil-containing sludge, which were used to evaluate its potential for the production of carbon products in the chemical activation process. Obviously, the as-synthesized sample still contained large non-combustible fractions, including a significant moisture content (65.44 wt%) and ash content (24.98 wt%). It showed high separation efficiency for residual oil in the waste oil recycling plant, resulting in only about 10 wt% of combustible material remaining in the oil-containing sludge. Using the preliminary analysis conducted through EDS, Figure 1 shows the elemental compositions on the surface of the dried oil-containing sludge, indicating high carbon and oxygen contents (38.8 wt% and 29.0 wt%, respectively) and the presence of other inorganic elements like iron (19.5 wt%), silicon (6.9 wt%), magnesium (4.5 wt%), and sulfur (1.3 wt%). In this regard, the origin of oil-containing sludge could be derived from waste oil at the bottom of iron-made storage tanks in petroleum refinery plants.

Figure 2 depicts the results of the thermogravimetric analysis (TGA) and derivative thermogravimetry (DTG) measurements for the dried oil-containing sludge in the temperature range of 25 °C to 900 °C at four heating rates (5, 10, 15 and 20 °C/min) under an inert nitrogen atmosphere. These curves were normalized by the initial sample mass. Obviously, a significant weight loss of about 25–30% was observed in the temperature range of 200 °C to 500 °C. This can be attributed to the evaporation loss of residual heavy oil fractions still contained in the dried sludge.

### 3.2. Characterization of Resulting Carbon Products

The data on the pore properties of the resulting carbon products, including BET surface area (S_BET_), micropore surface area (S_micro_), external surface area (S_ext_), total pore volume (V_t_), micropore volume (V_micro_), and average pore diameter (D_av_), are summarized in Table 2. Using the BET surface area values as a porosity criterion, the main results can be concluded as follows:
The H_3_PO_4_ impregnation ratio had a positive impact on the pore properties of the resulting carbon products. The BET surface area indicated a slight increasing trend as the impregnation ratio increased. As compared to the carbon product without impregnation by H_3_PO_4_ (i.e., C-0-SL-600-30), the enhancement in pore development with impregnation seemed be insignificant, thus showing that its BET surface area was slightly higher than that of the other carbon product (i.e., C-05-SL-600-30).The pore properties of carbon products can be enhanced by post-washing with water and dilute acid solution; that is, the following specific surface areas were obtained: 50.799 and 49.462 m^2^/g (C-0-SL-600-30 and C-05-SL-600-30, respectively), 58.980 and 57.832 m^2^/g (C-0-SL-600-30-W and C-05-SL-600-30-W, respectively), and 63.132 and 69.132 m^2^/g (C-0-SL-600-30-A and C-05-SL-600-30-A, respectively). On the other hand, acid-washing resulted in better pore properties than those obtained by water-washing, possibly due to more dissolution (leaching-off) of inorganic minerals and more pores being developed and formed [30,31,34].

Figure 3 and Figure 4 showed the N_2_ adsorption and desorption isotherms and the pore size distributions for the carbon products without H_3_PO_4_ impregnation and with H_3_PO_4_ impregnation (using the lowest mass ratio of 0.5), respectively. The isotherms featured a hysteresis loop and were type IV isotherms, indicating mesoporous structures based on the classification of the International Union of Pure and Applied Chemistry (IUPAC) [32,33]. As seen in Figure 4, the mesopore size range was concentrated between 3.0 and 4.5 nm, which was obtained by using the data on the branches of the desorption isotherms and the Barrett–Joyner–Halenda (BJH) model [33].

The SEM images (3000 magnification) of the dried oil-containing sludge (SL) and the optimal carbon product (i.e., C-05-SL-600-30-A) are shown in Figure 5. The former seemed to be only slightly porous and heterogeneous (Figure 5a) due to its small BET surface area (3.795 m^2^/g). In contrast, the carbon product (Figure 5b) indicated a porous texture on its surface, thus having a higher BET surface area. In addition, the elemental compositions on the surfaces of the carbon products (i.e., C-05-SL-600-30-A) can be seen in Figure 6. Here, the significant peak on the far right is the background gold (Au) used to provide a conductive film on the surface. The contents of carbon, oxygen, silicon, magnesium, and iron were 24.2, 15.8, 5.0, 2.3, and 52.6 wt%, respectively. These results are consistent with a previous study [10].

## 4. Discussion

### 4.1. Thermochemical Characterization of Oil-Containing Sludge

As seen in Figure 1, the unidentified peak at around 2.2 keV was assigned to gold (Au) because this element was used to provide a conductive film on the surface through ion sputtering. Furthermore, the calorific value of the dried oil-containing sludge was only 13.06 MJ/kg. This value is significantly lower than that of coal and petroleum-based fuels [35,36]. Although oil-containing sludge can be treated with a combustion method, the emissions of particulates and sulfur oxides (SOx) from incineration plants or industrial boilers should be controlled by air pollution control systems like baghouses and flue-gas desulfurization (FGD). On the other hand, the value of about 25–30% obtained from the TGA curves (Figure 2) is close to the data obtained from the proximate analysis in Table 1 (i.e., 9.58 wt%/(1–0.6544) = 27.7 wt%). As shown in Figure 2, the weight loss at high temperatures (>900 °C) may be caused by the evaporation loss of inorganic fractions (or minerals) with low melting points.

### 4.2. Characterization of Resulting Carbon Products

As compared to previous studies on the production of carbon products from oil-containing sludge [10,11], the results of H_3_PO_4_ chemical activation at 600 °C in this work were superior to those obtained from CO_2_ physical activation at 850 °C for the same length of time (30 min). The maximal BET surface area value (69.13 m^2^/g, seen in Table 2) for the carbon product C-05-SL-600-30-A was larger than that (40.53 m^2^/g) obtained from CO_2_ physical activation and post-acid-washing [10]. Therefore, the resulting carbon product featured a hysteresis loop with type IV isotherms (as shown in Figure 3) and also indicated a porous texture on the surface (as shown in Figure 4 and Figure 5), thus having a higher BET surface area. In comparison to the EDS data on the initial sludge (as shown in Figure 1), the contents of carbon, silicon, and magnesium (identified in Figure 6) were slightly reduced, while the iron content increased from 19.5 wt% to 52.6 wt%. As mentioned above, carbon products with mesoporous and magnetic properties may be used as efficient adsorbents or catalyst supports. When disposed of or exhausted, they can easily be collected or separated by applying external magnets [37,38].

## 5. Conclusions

All kinds of waste oils, including engine oil, lubricating oil, cutting oil, and tank-bottom oil, have been recycled by waste management plants. However, they still generate large amounts of residual sludge containing oils and inorganic minerals (e.g., iron oxides). In the present study, a phosphoric acid (H_3_PO_4_) activation method was used to produce porous carbon products from oil-containing sludge. From the data on the pore properties of the resulting products under various process conditions, it was found that the H_3_PO_4_ impregnation ratio had a positive impact on pore development or formation. The higher the impregnation ratio, the better the pore properties. As predicted, the enhancement in the pore properties of the resulting carbon products from acid-washing was superior to that from water-washing and no post-washing. This could be ascribed to the leaching removal of residual H_3_PO_4_ and inherent minerals. The carbon product with the maximal BET surface area (69.132 m^2^/g) could be produced at a lower temperature (600 °C). More significantly, the resulting carbon product was a mesoporous and magnetic material, suggesting that it can be further used as an efficient adsorbent or catalyst support due to its simple recovery (or separation) when exhausted.

## Figures and Tables

**Figure 1 materials-17-06161-f001:**
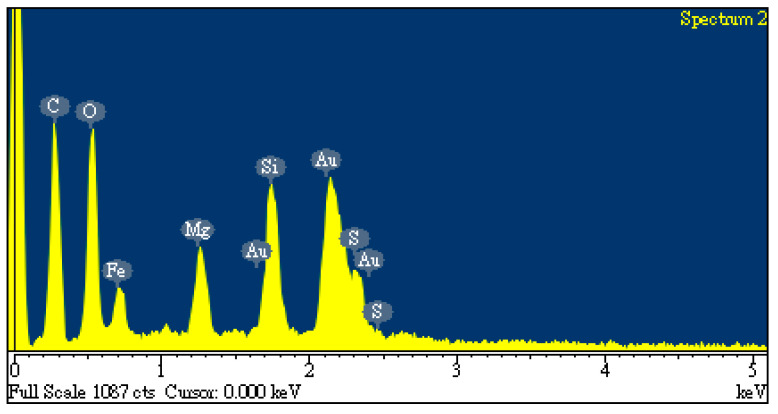
EDS spectra of oil-containing sludge.

**Figure 2 materials-17-06161-f002:**
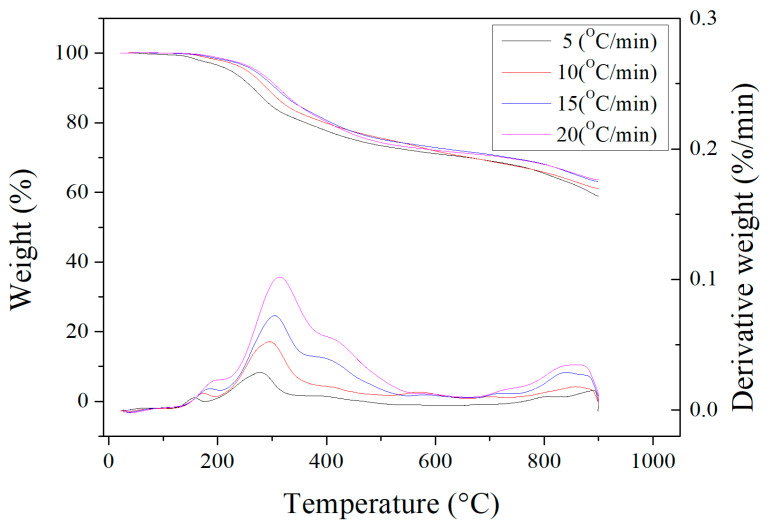
Thermogravimetric analysis (upper, shown by the vertical coordinates on the left side) and derivative thermogravimetry (lower, shown by the vertical coordinates on the right side) curves of dried oil-containing sludge.

**Figure 3 materials-17-06161-f003:**
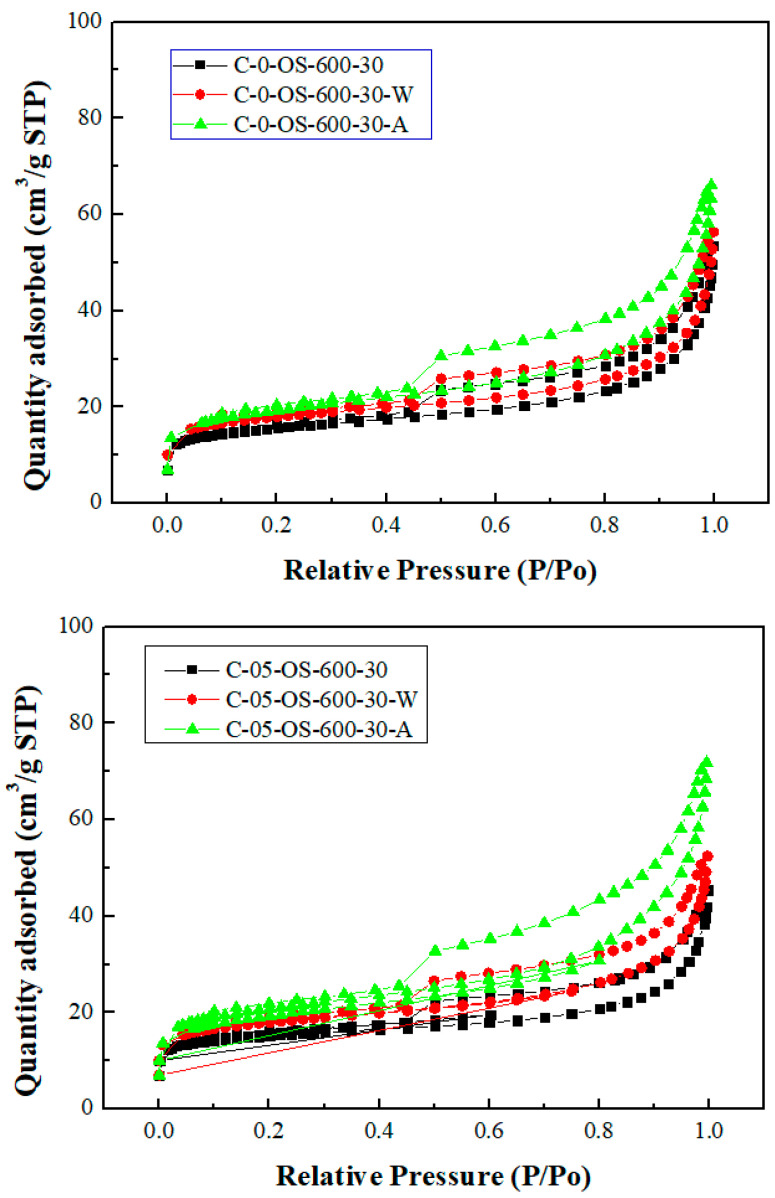
Nitrogen adsorption–desorption isotherms of carbon products. Upper images for no impregnation by H_3_PO_4_; lower images for impregnation ratio of 0.5.

**Figure 4 materials-17-06161-f004:**
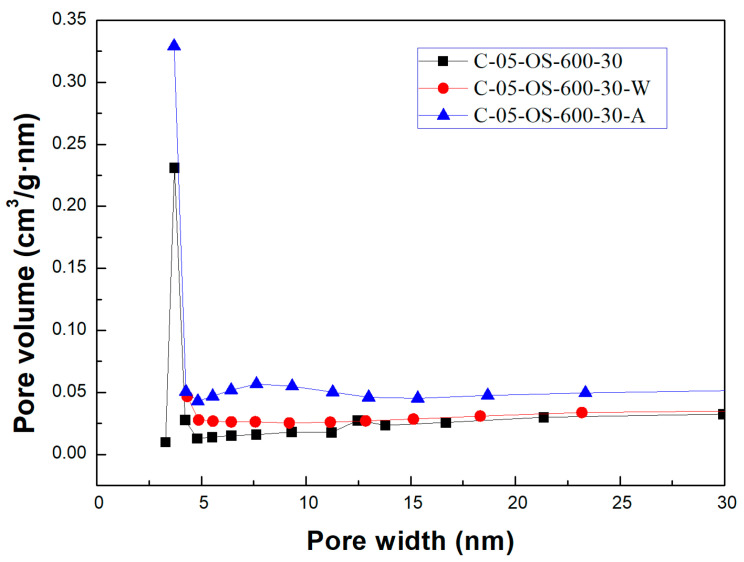
Pore size distributions of carbon products for impregnation ratio of 0.5.

**Figure 5 materials-17-06161-f005:**
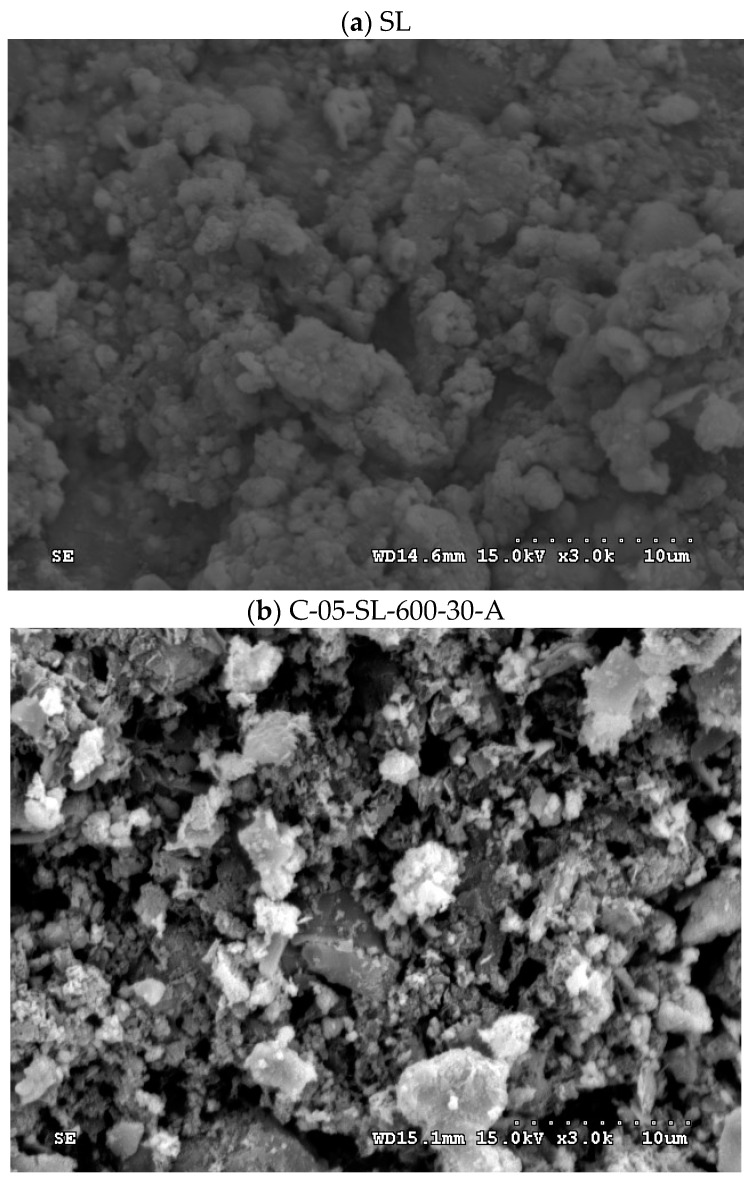
SEM images (×3000) of (**a**) oil-containing sludge (SL) and (**b**) resulting carbon product (C-05-SL-600-30-A).

**Figure 6 materials-17-06161-f006:**
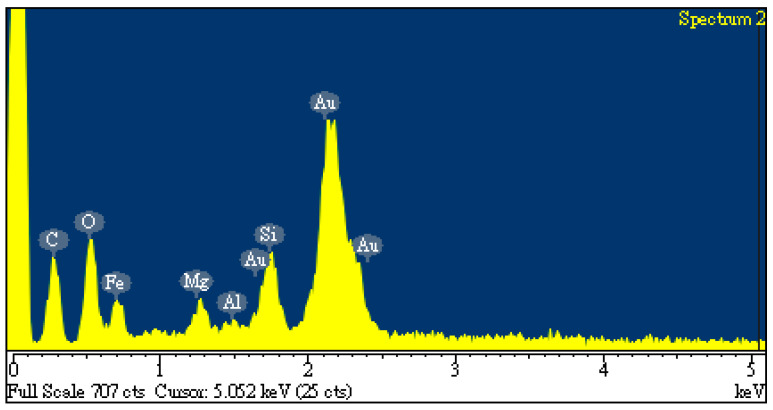
EDS spectra of resulting carbon product (C-05-SL-600-30-A).

**Table 1 materials-17-06161-t001:** Proximate analysis and calorific value of oil-containing sludge.

Property ^a^	Value
Proximate analysis ^b^	
Moisture (wt%)	65.44 ± 0.34
Ash (wt%)	24.98 ± 0.47
Combustible fraction (wt%)	9.58 ± 0.35
Calorific value (MJ/kg) ^c^	13.06 ± 0.92

^a^ The mean ± standard deviation for three determinations. ^b^ As-synthesized sample. ^c^ Dry basis.

**Table 2 materials-17-06161-t002:** Pore properties of resulting carbon products ^a^.

Code of Carbon Product	S_BET_ ^b^ (m^2^/g)	S_micro_ ^c^ (m^2^/g)	S_ext_ ^d^ (m^2^/g)	V_t_ ^e^ (cm^3^/g)	V_micro_ ^c^ (cm^3^/g)	D_av_ ^f^ (nm)
C-0-SL-600-30	50.799	24.947	25.852	0.0769	0.0130	6.053
C-05-SL-600-30	49.462	29.524	19.938	0.0657	0.0148	5.314
C-10-SL-600-30	51.610	31.352	20.258	0.0613	0.0163	4.751
C-15-SL-600-30	54.563	39.245	15.318	0.0566	0.0197	4.147
C-20-SL-600-30	58.611	38.529	20.082	0.0590	0.0195	4.028
C-0-SL-600-30-W	58.980	31.315	27.665	0.0796	0.0157	5.399
C-05-SL-600-30-W	57.832	29.589	25.243	0.0760	0.0598	5.256
C-10-SL-600-30-W	60.168	36.468	23.700	0.0674	0.0190	4.478
C-15-SL-600-30-W	57.255	37.268	19.987	0.0584	0.0187	4.083
C-20-SL-600-30-W	59.296	37.800	21.496	0.0686	0.0197	4.629
C-0-SL-600-30-A	63.132	26.641	36.6889	0.0979	0.0138	6.185
C-05-SL-600-30-A	69.132	31.152	37.980	0.1100	0.0155	6.365

^a^ The carbon products were produced from oil-containing sludge (SL) with different impregnation ratios (H_3_PO_4_/SL) at 600 °C for 30 min. ^b^ BET surface area was correlated in a relative pressure range of 0.06–0.30. ^c^ Obtained by the *t*-plot method. ^d^ Obtained by deducting the micropore surface area (S_micro_) from the BET surface area (S_BET_). ^e^ Total pore volume was obtained at a relative pressure of 0.995. ^f^ Obtained from the ratio of the total pore volume (V_t_) to the BET surface area (S_BET_) (i.e., Average pore width = 4 × V_t_/S_BET_).

## Data Availability

The data presented in this study are available in [https://world-nuclear.org/information-library/facts-and-figures/heat-values-of-various-fuels] (accessed on 14 November 2024).

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
