# Peer review of "Conversion of Oil-Containing Residue from Waste Oil Recycling Plant into Porous Carbon Materials Through Activation Method with Phosphoric Acid"

_materials, 2024, doi:10.3390/ma17246161_

Round 1
Reviewer 1 Report
Comments and Suggestions for Authors
Regarding the work "Conversion of oil-containing residue from waste oil recycling plant into porous carbon materials by the activation method with phosphoric acid".
In “Figure 1. EDS spectra of oil-containing sludge” although it mentions that the coating was with gold, it would be convenient for the EDS image to be complete. To identify the main elements present.
Figure 5. SEM images (×1,000) of (a) oil-containing sludge (SL) and (b) resulting carbon product (C-05-SL-600-30-A), is suggested to describe the images in the text; What would its morphology be like? In addition, the same image shows where the analysis was carried out on the particle that would help locate the carbon, oxygen, silicon, magnesium and iron, and possibly with a small table attached. In this context, it is recommended to include an image with a higher magnification to better appreciate the morphology of your material.
In “Fig. 6. EDS spectra of resulting carbon product (C-05-SL-600-30-A)” is similar to one, it would be convenient for the EDS image to be complete.
It is suggested that in section “3. Results and discussion” separates the discussion to make it clearer.
The conclusions should be clear and support the results, so it is suggested that you could possibly separate the section into concise paragraphs.
Author Response
Point 1: In “Figure 1. EDS spectra of oil-containing sludge” although it mentions that the coating was with gold, it would be convenient for the EDS image to be complete. To identify the main elements present.
Response 1: As pointed out by the reviewer, the unidentified peak at around 2.2 keV was assigned to the gold (Au) in Figure 1.
Point 2: Figure 5. SEM images (×1,000) of (a) oil-containing sludge (SL) and (b) resulting carbon product (C-05-SL-600-30-A), is suggested to describe the images in the text; What would its morphology be like? In addition, the same image shows where the analysis was carried out on the particle that would help locate the carbon, oxygen, silicon, magnesium and iron, and possibly with a small table attached. In this context, it is recommended to include an image with a higher magnification to better appreciate the morphology of your material.
Response 2: As pointed out by the reviewer, the SEM images (×3,000) of (a) oil-containing sludge (SL) and (b) resulting carbon product (C-05-SL-600-30-A) were adopted to make them more clear.
Point 3: In “Fig. 6. EDS spectra of resulting carbon product (C-05-SL-600-30-A)” is similar to one, it would be convenient for the EDS image to be complete.
Response 3: As pointed out by the reviewer, the unidentified peak at around 2.2 keV was assigned to the gold (Au) in Figure 6.
Point 4: It is suggested that in section “3. Results and discussion” separates the discussion to make it clearer.
Response 4: As pointed out by the reviewer, the Discussion has been separated from the section “3. Results and discussion”, making it clearer.
Point 5: The conclusions should be clear and support the results, so it is suggested that you could possibly separate the section into concise paragraphs.
Response 5: As suggested by the reviewer, the section (“Conclusions”) was rewritten to make it clear and concise.
Reviewer 2 Report
Comments and Suggestions for Authors
The article is very concise, which is rather helpful. It presents a topic which obviously is not new, but the authors concentrate on their specific material and thus present relevant data.
Nevertheless, there are some flaws in the presentation and in the language, which should be corrected.
l 15 "are" instead of "were"
l 38 what power? incomprehensible part of the sentence
l 39 Mn is not Mg - this MUST be corrected
l 109 flow rate set TO 50
l 111 XRF for surface analysis? This method goes some way into the sample, so it is a bit bold to describe it as a surface analysis method.
l 115 it IS generally
l 117 "acting" instead of "acted"
l 182 the authors state a rather low heating value. Well, with 65% moisture and 25% ash - where should the energy come from? This really is no surprise.
l 221 I wonder if the authors really did BET down to that 3-digit precision. It rather looks like data simply taken from the machine.
l 258 the argument is the wrong way round. BET is the result of low porosity and heterogeneity (thus the latter does not necessarily apply) and not their reason.
l 264 again here, XRF does not provide such precise data down to 2 digits.
Author Response
Point 1: L 15 "are" instead of "were".
Response 1: As pointed out by the reviewer, the word has been used in the revised manuscript.
Point 2: L 38 what power? incomprehensible part of the sentence.
Response 2: As pointed out by the reviewer, the word has been corrected in the revised manuscript.
Point 3: L 39 Mn is not Mg - this MUST be corrected.
Response 3: As pointed out by the reviewer, the word has been corrected in the revised manuscript.
Point 4: L 109 flow rate set TO 50.
Response 4: As pointed out by the reviewer, the word has been corrected in the revised manuscript.
Point 5: L 111 XRF for surface analysis? This method goes some way into the sample, so it is a bit bold to describe it as a surface analysis method.
Response 5: In this work, EDS (energy-dispersive X-ray spectroscopy, not XRF) was used to determine the elemental compositions on the sample surface.
Point 6: L 115 it IS generally.
Response 6: As pointed out by the reviewer, the sentence has been corrected in the revised manuscript.
Point 7: L 117 "acting" instead of "acted".
Response 7: As pointed out by the reviewer, the word has been corrected in the revised manuscript.
Point 8: L 182 the authors state a rather low heating value. Well, with 65% moisture and 25% ash - where should the energy come from? This really is no surprise.
Response 8: As pointed out by the reviewer, the inconsistent data on proximate analysis and calorific value were due to different samples. The former data were based on as-received sample, but the latter value was determined by its dried sample.
Point 9: L 221 I wonder if the authors really did BET down to that 3-digit precision. It rather looks like data simply taken from the machine.
Response 9: Actually, the original data on BET surface area showed 4-digit precision from the machine system. The data with 3-digit precision were adopted by rounding.
Point 10: L 258 the argument is the wrong way round. BET is the result of low porosity and heterogeneity (thus the latter does not necessarily apply) and not their reason.
Response 10: As pointed out by the reviewer, the sentence has been revised to make it reasonable.
Point 11: L 264 again here, XRF does not provide such precise data down to 2 digits.
Response 11: As pointed out by the reviewer, the data on elemental compositions with 1-digit were adopted by rounding.